# Prevalence and determinants of non-fistulous urinary incontinence among Ghanaian women seeking gynaecologic care at a teaching hospital

Anthony Amanfo Ofori[1]*, Joseph Osarfo[2], Evans Kofi Agbeno[1], Wisdom Klutse Azanu[3], Henry Sakyi Opare-Addo[4]

1 Department of Obstetrics and Gynecology, School of Medical Sciences, University of Cape Coast, Cape Coast, Ghana, 2 Effiduase District Hospital, Ghana Health Service, Ashanti Region, Ghana, 3 Department of Obstetrics and Gynecology, School of Medical Sciences, University of Health and Allied Science, Ho, Ghana, 4 Department of Obstetrics and Gynecology, School of Medical Sciences, Kwame Nkrumah University of Science and Technology, Kumasi, Ghana

* ofooh@yahoo.com, anthony.ofori@ucc.edu.gh

**Data Availability Statement:** All relevant data are within the manuscript and its Supporting Information files.

## Abstract

The study assessed the prevalence and determinants of non-fistulous urinary incontinence among gynaecologic care seekers as well as its interference with everyday life activities of affected women. A cross-sectional study involving 400 women was conducted in a tertiary facility in Ghana. Urinary incontinence was assessed using the International Consultation on Incontinence Questionnaire-short form (ICIQ-SF) which has not been validated locally. The questionnaire was administered mostly in the Asante Twi language with translation done at the time of the interview. The data was analysed for proportions and associations between selected variables. The prevalence of urinary incontinence was 12%, the common types being urgency (33.3%), stress (22.9%), and mixed (20.8%). Age $\geq$60 years compared to 18–39 years (OR 3.66 95%CI 1.48–9.00 P = 0.005), and a history of chronic cough (OR 3.80 95% CI 1.36–10.58 P = 0.01) were associated with urinary incontinence. Women with education beyond the basic level were 72% less likely to experience urinary incontinence (OR 0.28 95%CI 0.08–0.96 P = 0.04). Urinary incontinence interferes with everyday life activities of most affected women. Non-fistulous urinary incontinence is relatively common among gynaecologic care seekers yet very few women were referred with such a diagnosis. Advocacy measures aimed at urging affected women to report the condition and educating the general population on potential causes, prevention and treatment are needed.

## Introduction

Urinary incontinence, defined by the International Continence Society as a complaint of any involuntary leakage of urine [1] comprises two main types; urethral (non-fistulous) and extra urethral (fistulous). It is a common problem with a global prevalence of 4.8%-54.8% [2] and

**Funding:** the authors received no specific funding for this work.

**Competing interests:** the authors have declared that no competing interests exist.

negative effects on quality of life with respect to social life, personal relationship, feelings, sleep and energy [3, 4]. Coyne and colleagues [5] reported estimated costs of overactive bladder with urgency urinary incontinence of US$65.9 billion in 2007 with projected cost of US$82.6 billion in 2020 in the United States.

The wide reported prevalence stems from differences in definitions, target populations, and methodology in various studies [2]. Studies with broad definition of incontinence such as "any loss of urine in the past 12 months" had higher prevalence than those defining urinary incontinence over a shorter period of time such as "two or more bedwetting episodes in the past month" [2].

Age, race, obesity, parity, previous hysterectomy, smoking, alcohol consumption, chronic cough, chronic constipation, assisted delivery and other characteristics and practices have been reported to be associated with an increased occurrence of urinary incontinence [2, 6–11].

The three main subtypes of urinary incontinence in women are stress, urgency and mixed urinary incontinence. For the population as a whole stress urinary incontinence is the most common among the three subtypes. The prevalence of stress incontinence however peaks around the fifth decade of life and thereafter the prevalence of urgency and mixed incontinence continue to increase with mixed incontinence being the most prevalent subtype in older women [12, 13].

Non-fistulous urinary incontinence has been studied extensively in Europe, China and United States of America [2, 6–9, 14–16].

Similarly, prevalence of 5.2%-39% has been reported in some African studies [17–20].

In Ghana, the concern has been on urinary incontinence from obstetric fistula [21, 22] and very little research has been done on the subject of non-fistulous incontinence. Literature search revealed only one published study in Ghana by Adanu et al who reported a 22.5% self-reported prevalence and 41.5% prevalence for demonstrable stress incontinence among women with full bladder at an ultrasound clinic [23]. The higher prevalence of demonstrable stress incontinence may be an overestimate as women with an uncomfortably full bladder may leak urine on coughing even though that might not be the situation under normal conditions. It may also be that women do not often report the condition because it is not deemed worrisome or out of embarrassment [19]. In addition, the study did not report on whether incontinence affected the quality of life of participants.

The paucity of epidemiologic data on non-fistulous incontinence in Ghana potentially creates the impression the condition is not a problem in the country. The result is that little or no advocacy is done about non-fistulous incontinence. Women are not educated about the causes and treatment options available to them. Many clinicians are not familiar with current information on appropriate evaluation and treatment of urinary incontinence and treatment centres lack the necessary resources to effectively manage affected women.

An assessment of the prevalence and determinants of non-fistulous urinary incontinence as well as its self-reported interference with everyday life activities of those affected was conducted among women visiting an out-patient gynaecology clinic at a tertiary facility in the middle belt of Ghana.

## Materials and methods

An analytical cross-sectional study was conducted among women aged ≥18 years who accessed care at the gynaecology out-patient clinic of the Komfo Anokye Teaching Hospital in Kumasi between 1st January and 31st March 2015. Total attendance to the gynaecology clinic in 2013 was 7000 (KATH, Biostatistics Unit). Komfo Anokye Teaching Hospital is the second

largest teaching hospital in Ghana and a major referral centre for hospitals in the central and northern parts of Ghana

The sample size, N, was estimated using the formula $N = Z^2pq/d^2$ where;

Z is reliability coefficient (at 95% confidence interval and 5% level of significance, Z is 1.96),

p is the population proportion of the factor under investigation; in this study p is the prevalence of urinary incontinence;

q = 1-p and

d is the desired difference between the actual population proportion and what the study will realize.

Assuming the global prevalence of 54.8% [2]; at a desired difference of 5%, a sample size of 381 was estimated as follows $N = 1.96^2 \times 0.54 \times 0.46 / 0.05^2 = 381$. This figure was approximated to 400 assuming a non-response rate of 5%.

A study in Ghana [23] reported a 41.5% prevalence of demonstrable stress urinary incontinence. However, a high-end global prevalence of 54.8% was chosen to allow for a conservative overestimation to ensure adequate sampling.

Between 1[st] January 2015 and 31[st] March 2015, all women accessing care at the gynaecology clinic irrespective of presenting complaints were assessed for eligibility by research assistants who were midwives working at the gynaecology clinic. Women below 18 years of age, women in their puerperium and those with a referral diagnosis of fistulous incontinence were excluded from the study at this point. The objectives of the study were explained to those eligible to partake in the study prior to consulting their physicians by research assistants who had been trained to administer the questionnaires. The vast majority of questionnaires were administered through verbal translation into the local Asante Twi language as most participants had little or no English reading and comprehension skills while only a handful were self-administered in English. Translation of the questionnaire into the local language was done at the same time of the interview. Written informed consent were obtained from those willing to participate in the study.

Inclusion criteria comprised all women who accessed care at the gynaecology clinic irrespective of presenting complaints during the study period and consented to participate in the study.

Exclusion criteria comprised i) women declining to give consent ii) women below 18 years of age iii) women with confirmed pregnancy during the study period iv) women in their puerperium during the study period v) women with a referral diagnosis of fistulous incontinence during the study period vi) women responding 'NO' to the screening question "have you experienced involuntary leakage of urine in the past one month" were excluded from completing the ICIQ-SF(the second part of the questionnaire).

Data was collected using an orally-administered two-part questionnaire. Part one was for socio-demographic data (age, educational level, marital status, occupation), reasons for referral, general medical history and health habits (Body Mass Index, diabetes mellitus, hypertension, chronic cough, chronic constipation, smoking, consumption of alcohol, tea/coffee and carbonated drinks), and obstetric and gynaecologic history (gravidity, parity, mode of delivery, perineal injury, menopausal status, history of pelvic surgery, history of vaginal surgery). These variables were studied because of their reported association with non-fistulous incontinence [2, 6–9].

The second part of the questionnaire was made up of the International Consultation on Incontinence Questionnaire–Short Form (ICIQ-SF) [24]. This symptom questionnaire is made up of three scored questions regarding frequency of urinary incontinence (scored from 0 to 5), amount of leakage (scored from 0 to 6), and overall inconvenience (scored from 0 to 10). These three items sum up to give an overall score from 0 to 21 points, which has been

graded by Klovning A et al. [25] as slight (1–5), moderate (6–12), severe (13–18), and very severe (19–21). A score of 0 indicates no incontinence symptoms. A fourth question of the ICIQ-SF, designed to elucidate type of urinary incontinence, is unscored. A screening question "have you experienced any involuntary loss of urine in the past one month?" was asked prior to completing the ICIQ-SF. Responders answering "yes" to the screening question proceeded to answer the four items on the ICIQ-SF. Those answering "no" to the scrrening question did not complete the ICIQ-SF. The screening question was added in order not to burden responders without incontinence with the task of completing the ICIQ-SF.The ICIQ-SF, as used, has not been validated in the Ghanaian population.

Data was double-entered in SPSS version 20 (IBM, Armonk, NY, USA) and analysed using Stata 13 (Stata Corp, College Station, Texas, USA) after consistency checks. The data was summarized using frequencies and means with standard deviation and range

Associations between urinary incontinence and exposure variables were analyzed using chi-square test for categorical data and t-test for continuous data. Variables found to be statistically significant in the univariate analysis (i.e. $p < 0.1$) were included in a multivariate model to explore their independent associations with incontinence status and adjusted odds ratios with 95% confidence intervals estimated. All exposures with p value $<0.05$ in the multivariate analysis were considered significant.

The study was approved by the committee for Human Research, Publication and Ethics (CHRPE) of the Kwame Nkrumah University of Science and Technology (CHRPE/AP/239/14). Consent was also sought and obtained from Komfo Anokye Teaching Hospital's administration. Written Informed consent was obtained from all eligible participants. All clients were assigned a study number and no personal identifiers were used.

## Results and discussions

### Results

Between 1st January 2015 and 31st March 2015, a total of 748 women were assessed for study eligibility. Of the 438 women deemed eligible, 400 were recruited into the study (see Fig 1).

Table 1 shows the background characteristics of the study women. The mean age of respondents was 42.7 ±12.5 (mean ± SD) with a range of 19 to 88 years. Over three quarters of the respondents (304/400, 76%) had up to basic level education defined as 9 years of primary school education and 12.8% (51/400) were farmers.

The most common referral diagnosis was fibroid uterus (103/400, 25.8%) (See Fig 2). Only 0.75% (3/400) of the study women reported with a referral diagnosis of urinary incontinence. However, with questionnaire administration, urinary incontinence prevalence was 12% (48/400) (95% CI 8.80%-15.19%). A third (16/48, 33.3%) of respondents with incontinence had urgency incontinence, 22.9% (11/48), had stress urinary incontinence and about a fifth (10/48, 20.8%) had mixed incontinence. Furthermore, more than a third of incontinent women (18/48, 38%) experienced leakage of urine all the time, 21% (10/48) leaked urine once a day, while 8% (4/48) leaked several times a day. The amount of leakage was reportedly small for half of them (24/48, 50%), moderate for a third (16/48, 33%) and large for the remaining (8/48, 17%).

On the overall interference with everyday life scale which ranged from 0 to 10 (with 0 representing no inconvenience and 10 representing a great deal of inconvenience) only 2% (1/48) of incontinent women reported no interference with daily life. Approximately 31% (15/48) reported mild interference with daily life whiles about 42% of incontinent women said urinary incontinence moderately interfered with their daily life. Interference of urinary incontinence with everyday life of affected women is shown in Table 2 below.

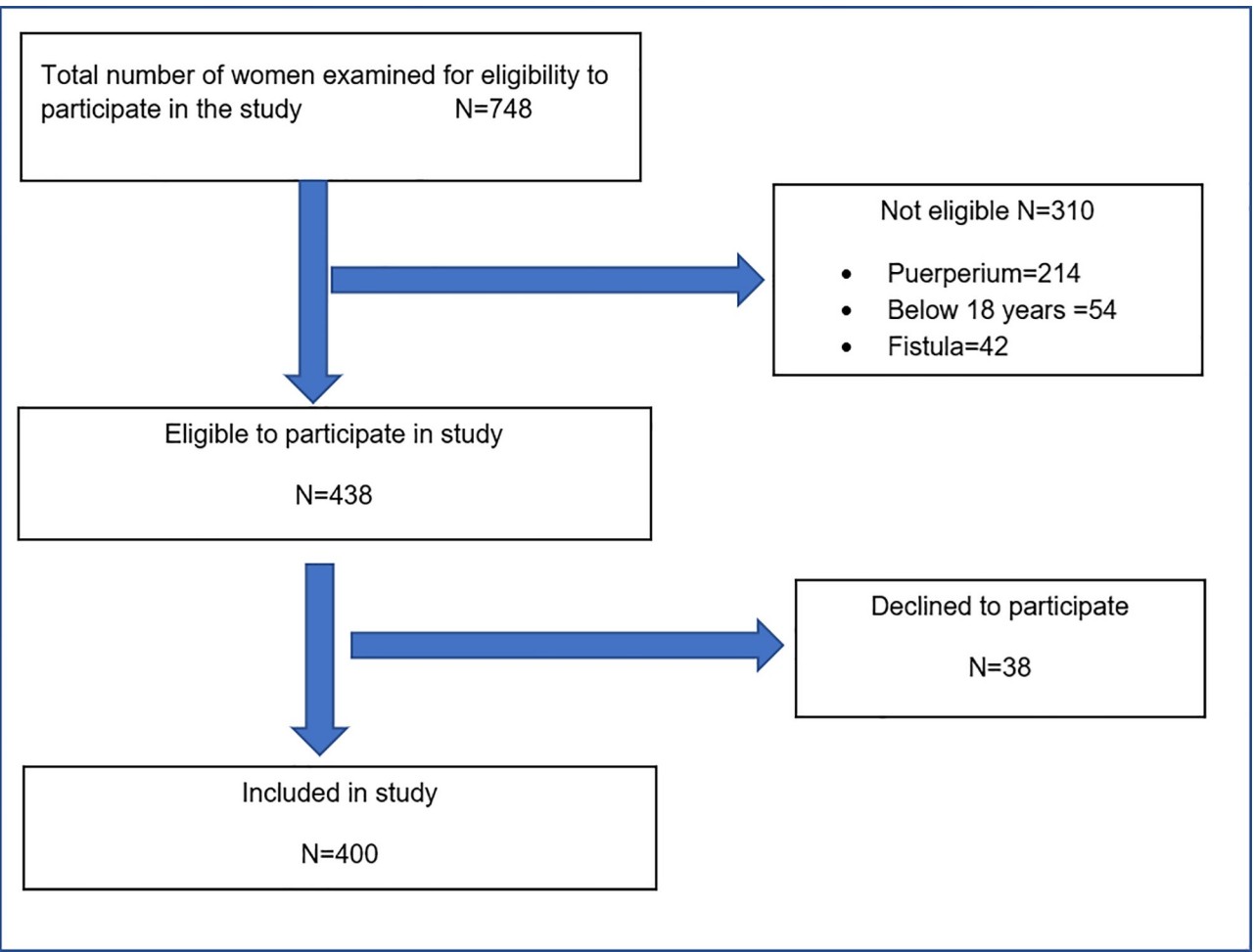

**Fig 1. Profile of participant recruitment.**

The mean ICIQ score of incontinent women was 11.60 ± 5.03 (mean ±SD) with a range of 3 to 21. The proportion of incontinent women by ICIQ score is shown in Fig 3.

In the univariate analysis (see Tables 1 and 3), age, level of education, marital status, occupation, BMI, gravidity, parity, number of vaginal deliveries, menopausal status, Diabetes Mellitus, chronic cough and smoking were significantly associated with urinary incontinence. Adjusting for the effect of other variables in the multivariate analysis, age (P = 0.02), history of chronic cough (P = 0.01) and educational status of women (P = 0.04) were associated with the occurrence of urinary incontinence.

Table 4 shows the crude and adjusted odds ratio for factors associated with incontinence in the final model. Women aged ≥60 years were three and half times more likely to experience urinary incontinence compared to women aged 18–39 years. (AOR = 3.65, 95% CI1.48–9.00 P = 0.005). Again, women with a history of chronic cough in the past one year had approximately four times the odds of urinary incontinence compared to women without such history (AOR = 3.80 95% CI 1.36–10.58, P = 0.01). Having education beyond the basic level was however protective against urinary incontinence (AOR = 0.27 95% CI 0.08–0.96 P = 0.04). Almost all incontinent women (47/48, 97.9%) reported urinary incontinence interfere with their daily activities.

**Table 1. Socio-demographic and obstetrics and gynaecologic characteristics of participants and their association with urinary incontinence.**

| Variable | Incontinence | | P-value |
|---|---|---|---|
| | No | Yes | |
| **Age in groups (years)** | N (%) | N (%) | <0.001 |
| 18–39 | 177(94.15) | 11 (5.85) | |
| 40–59 | 134(85.35) | 23(14.65) | |
| 60 and above | 41(74.55) | 14(25.45) | |
| **Educational status** | | | 0.002 |
| Up to basic education | 259(85.20) | 45(14.80) | |
| Beyond Basic education | 93(96.88) | 3(3.13) | |
| **Occupational status** | | | 0.013 |
| Unemployed | 54(87.10) | 8(12.90) | |
| Traders/artisans | 226(90.76) | 23(9.24) | |
| Civil/public servants | 34(89.47) | 4(10.53) | |
| Farmers | 38(74.51) | 13(25.49) | |
| **Marital status** | | | <0.001 |
| Single | 72(97.30) | 2(2.70) | |
| Married/cohabiting | 219(89.02) | 27(10.98) | |
| Divorced | 29(85.29) | 5(14.71) | |
| widow | 32(69.57) | 14(30.43) | |
| **Gravidity** | | | 0.001 |
| 0 | 74(94.87) | 4(5.13) | |
| 1–4 | 174(91.10) | 17(8.90) | |
| 5 or more | 104(79.39) | 27(20.61) | |
| **Parity** | | | <0.001 |
| 0 | 107(93.04) | 8(6.96) | |
| 1–4 | 169(92.35) | 14(7.65) | |
| 5 or more | 76(74.51) | 26(25.49) | |
| **Mode of delivery** | | | 0.312 |
| Only SVD | 196(84.48) | 36(15.52) | |
| Only C/S | 16(94.12) | 1(5.88) | |
| SVD +C/S | 33(91.67) | 3(8.33) | |
| **Number of vaginal deliveries** | | | <0.001 |
| 1 | 50(98.04) | 1(1.96) | |
| 2–4 | 110(89.43) | 13(10.57) | |
| 5 or more | 69(73.40) | 25(26.60) | |
| **Perineal injuries** | | | 0.174 |
| No injury | 208(89.66) | 24(10.34) | |
| Injury | 137(85.09) | 24(14.91) | |
| **Menopausal status** | | | <0.001 |
| No | 275(92.28) | 23(7.72) | |
| Yes | 77(75.49) | 25(24.51) | |
| **Hysterectomy** | | | 0.447 |
| No hysterectomy | 320(87.67) | 45(12.33) | |
| Had hysterectomy | 25(92.59) | 2(7.41) | |
| **Vaginal surgery** | | | 0.526 |
| No vaginal surgery | 334(88.13) | 45(11.87) | |
| Had vaginal surgery | 9(81.82) | 2(18.18) | |

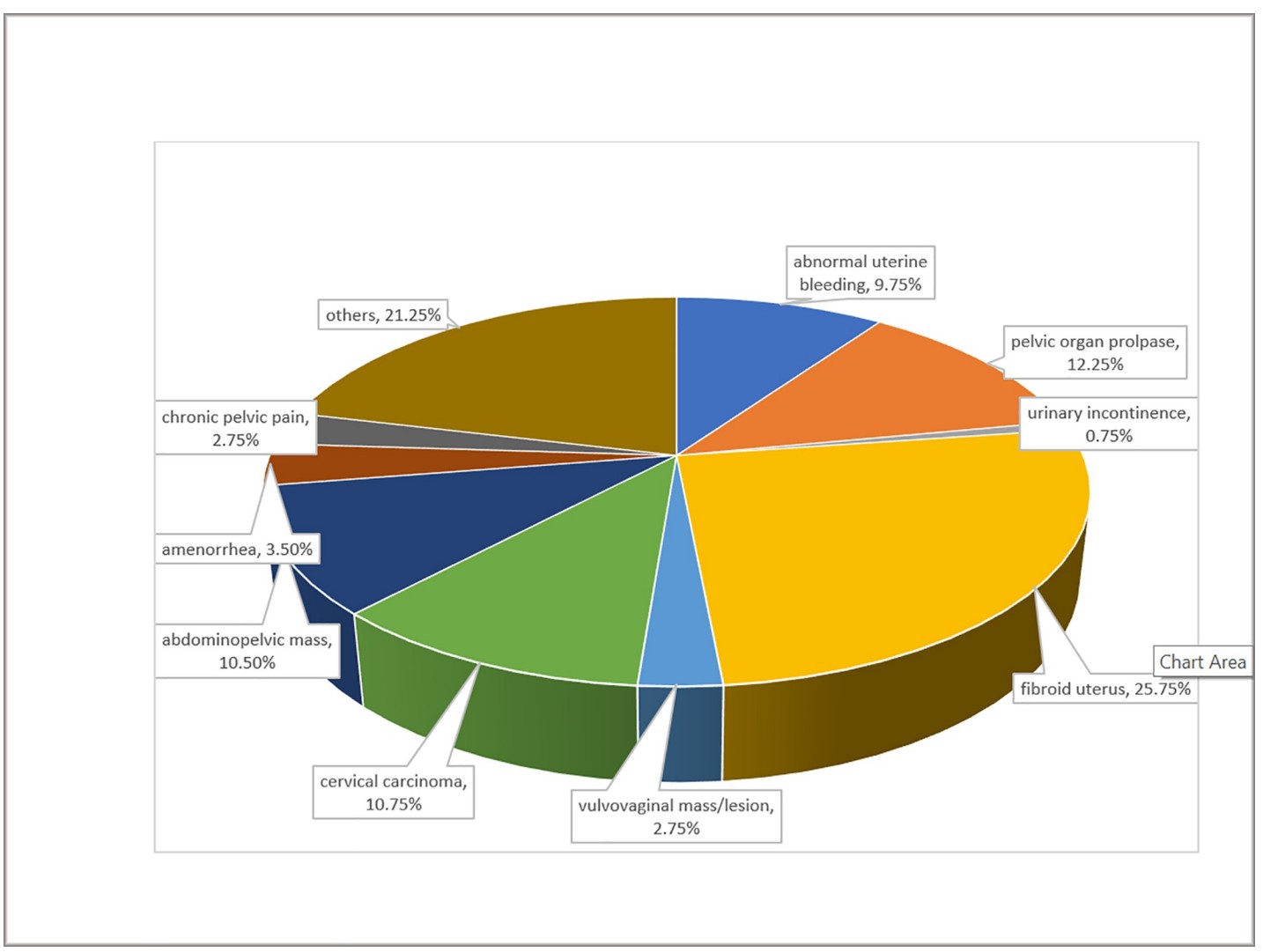

**Fig 2. Referral diagnoses of study participants (%).**

## Discussion

This cross-sectional study was conducted to assess the prevalence of urinary incontinence among women visiting an out-patient gynaecologic clinic, determine factors associated with its occurrence as well as its self-reported interference with everyday life activities of those affected. The overall prevalence of urinary incontinence was 12% though only 1% of study

**Table 2. Interference of urinary incontinence with everyday life of affected women.**

| ICIQ-SF interference with everyday life Score | N | % |
|---|---|---|
| 0 (not at all) | 1/48 | 2.08 |
| 1–3 (mild) | 15/48 | 31.25 |
| 4–6 (moderate) | 20/48 | 41.67 |
| 7–9 (severe) | 9/48 | 18.75 |
| 10 (great extent) | 3/48 | 6.25 |

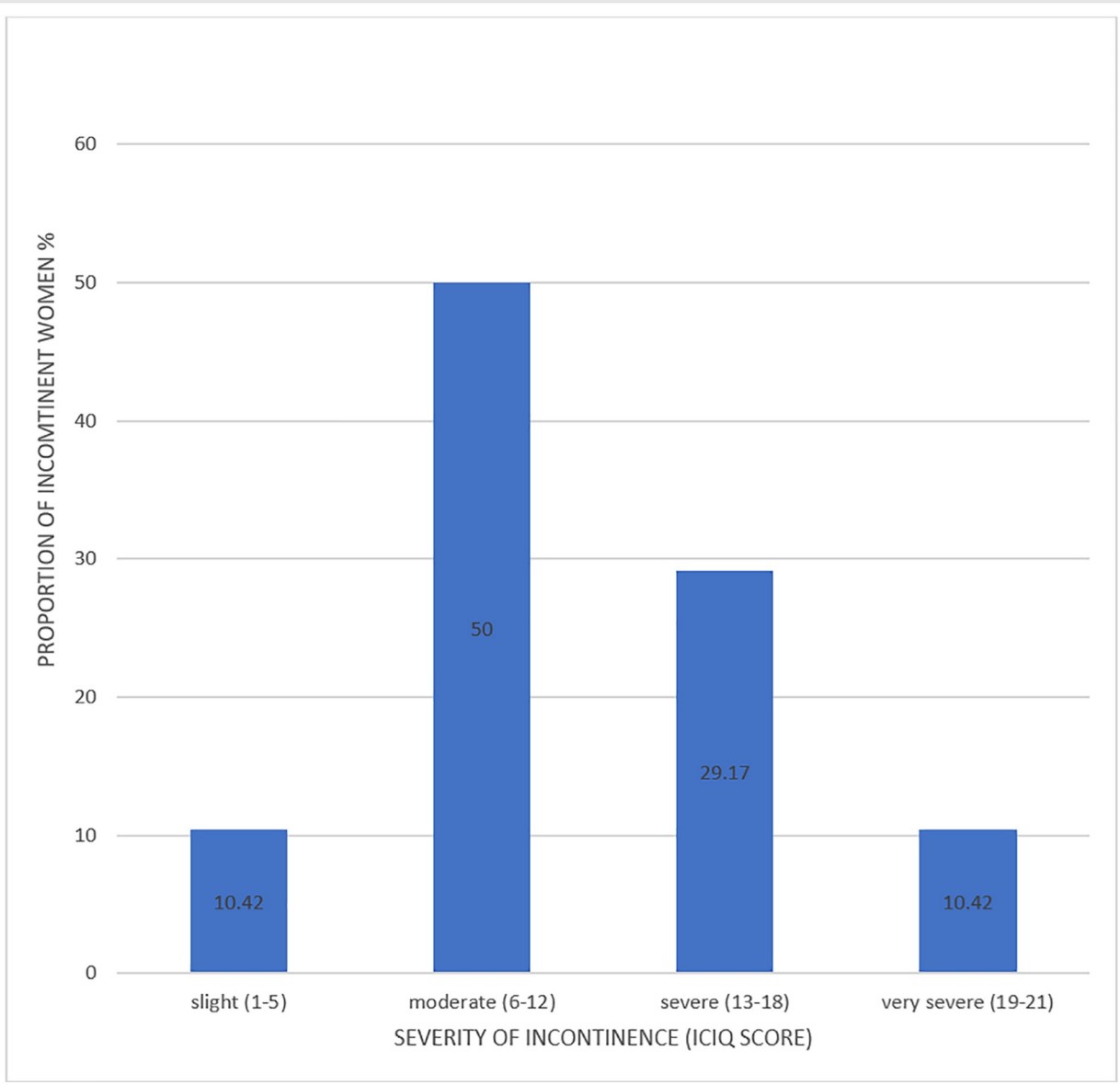

**Fig 3. Proportion of incontinent women by severity of incontinence (ICIQ score).**

population were referred with a diagnosis of urinary incontinence. Urgency incontinence was the commonest form of incontinence with most women grading their incontinence as either moderate or severe.

The overall prevalence of urinary incontinence of 12% in this study falls within the reported global prevalence of 4.8% - 58.4% [2] and is similar to the 12.2% [20] and 14.6% [8] prevalence reported among Nigerians and Black women in United States of America respectively. It was however lower compared to the 41.5%, prevalence rate of demonstrable stress urinary incontinence reported in another Ghanaian study [23] which required participants to present with a full bladder before the test. It is possible some respondents had an uncomfortably full bladder predisposing to leaks with cough (occurrence of demonstrable stress incontinence) even though that may not be routine.

**Table 3. Medical history and health habits of study women and their association with urinary incontinence.**

| Variables | Incontinence | | P-value |
|---|---|---|---|
| | **No** | **Yes** | |
| **Body mass index** | **N (%)** | **N (%)** | 0.054 |
| Non obese | 224(85.82) | 37(14.18) | |
| Obese[a] | 123(92.48) | 10(7.52) | |
| **Diabetes mellitus** | | | 0.041 |
| Yes | 27(77.14) | 8(22.86) | |
| No | 322(88.95) | 40(11.05) | |
| **Chronic constipation[b]** | | | 0.602 |
| Yes | 16(84.21) | 3(15.79) | |
| No | 336(88.19) | 45(11.81) | |
| **Chronic cough[c]** | | | 0.002 |
| Yes | 14(66.67) | 7(33.33) | |
| No | 338(89.18) | 41(10.82) | |
| **Smoking** | | | 0.066 |
| Yes | 15(75.00) | 5(25.00) | |
| No | 337(88.68) | 43(11.32) | |
| **Alcohol** | | | 0.205 |
| Yes | 37(82.22) | 8(17.78) | |
| No | 315(88.73) | 40(11.27) | |
| **Tea/coffee** | | | 0.639 |
| Nil/Occasionally ($\leq$ 1 cup /week) | 299(87.68) | 42(12.32) | |
| Frequently | 53(89.83) | 6(10.17) | |
| **Carbonated/fizzy drinks** | | | 0.839 |
| Nil/Occasionally ($\leq$ 1 bottle/week) | 335(87.93) | 46(12.07) | |
| Frequently | 17(89.47) | 2(10.53) | |

[a]Obese: BMI $\geq$30kg/m$^2$

[b]Chronic constipation: less than 3 bowel movements (stools) in a week for 6 months in the past one year

[c]Chronic cough: cough lasting 8 weeks or more in the last one year.

Again, the prevalence rate from the present study was also lower than the 21.4% [18] and 39% [19] reported in studies from Nigeria. The relatively narrow definition of urinary incontinence used in the present study compared to such definitions as "ever leaked urine in the past" and "had any leakage of urine occurred in the past one year" mostly used in these studies may account for the disparity. Assuming a common definition in studies assessing prevalence of non-fistulous urinary incontinence may be a first step towards obtaining a true global or regional prevalence. Such a common definition could be further modified to accommodate age and other appropriate differences.

Less than 1% of the study population had a referral diagnosis of incontinence. It is possible that women with incontinence did not report the condition to their primary physicians out of embarrassment or did not find their conditions bothersome or life threatening as previously reported [18, 26]. Some women may also consider the condition as part of aging, may be afraid of the complications of treatment, do not know what help is available or where to seek help [18, 26].

The common types of urinary incontinence recorded in this study were urgency (33.3%), stress (22.9%), and mixed (20.8%). This is consistent with findings from previous studies that identified stress, urgency and mixed incontinence as the three main subtypes of incontinence

**Table 4. Crude and adjusted odds ratios of factors associated with urinary incontinence.**

| Variable | Crude odds ratio | Adjusted odds ratio | 95%CI (adjusted odds ratio) | P-Value |
|---|---|---|---|---|
| **Age (years)** | | | | |
| 18–39 | | | | |
| 40–59 | 2.76 | 2.04 | 0.93–4.46 | 0.07 |
| $\geq$ 60 | 5.49 | 3.65 | 1.48–9.0 | 0.005 |
| **Chronic cough** | | | | |
| No | | | | |
| Yes | 4.12 | 3.80 | 1.36–10.58 | 0.01 |
| **Educational status** | | | | |
| $\leq$ Basic | | | | |
| >Basic | 0.19 | 0.28 | 0.08–0.96 | 0.04 |
| **Occupational status** | | | | |
| Unemployed | | | | |
| Traders/Artisans | 0.68 | 0.69 | 0.22–2.10 | 0.5 |
| Civil/Public servants | 0.79 | 0.64 | 0.11–3.64 | 0.6 |
| Farmers | 2.30 | 1.32 | 0.45–3.94 | 0.6 |
| **Marital status** | | | | |
| Single | | | | |
| Divorced | 6.20 | 2.0 | 0.27–14.77 | 0.49 |
| Married/cohabiting | 4.40 | 2.6 | 0.53–13.26 | 0.23 |
| Widow | 15.75 | 4.7 | 0.75–30.48 | 0.09 |
| **BMI** | | | | |
| Non-obese | | | | |
| Obese | 0.49 | 0.61 | 0.27–1.38 | 0.2 |
| **Parity** | | | | |
| 0 | | | | |
| 1–4 | 1.10 | 0.7 | 0.24–2.0 | 0.5 |
| $\geq$5 | 4.50 | 1.3 | 0.39–4.2 | 0.6 |
| **Menopause** | | | | |
| No | | | | |
| Yes | 3.8 | 1.57 | 0.62–4.01 | 0.33 |
| **Diabetes Mellitus** | | | | |
| No | | | | |
| Yes | 0.41 | 0.78 | 0.27–2.25 | 0.6 |
| **Smoking** | | | | |
| No | | | | |
| Yes | 0.38 | 0.39 | 0.08–1.72 | 0.2 |

in women [12, 13, 17–19]. Studies however differ on which type is the most prevalent. This study identifies urgency incontinence as the commonest type similar to findings of Badejoko et al. [17], but differs from other studies in the sub-region that identified stress incontinence as the most common [18, 19].

Almost all the incontinent women reported that urinary incontinence interfere with their everyday life activity with more than half of them giving a score of 5 or more as reported in other studies [27, 28]. The ICIQ-SF, though limited in its assessment of impact on quality of life associated with incontinence, may be taken as a proxy for assessment of the quality of life of affected women.

Women aged $\geq$60 years were three and half times more likely to experience urinary incontinence compared to women aged 18–39 years. This finding compares favourably with previous studies that found age associations with urinary incontinence [29, 30]. The aged may be prone to the development of urinary incontinence due to hypoestrogenism, decreased urethral closure pressure and the development of detrusor overactivity [31]. In addition impaired mobility and increased nocturnal urine production may contribute to the development of urinary incontinence [31]. Some studies however did not find age to be significantly associated with urinary incontinence [27, 32].

The study found women with a history of chronic cough in the past year were about four times more likely to experience urinary incontinence compared to women without such a history and was consistent with previous reports [26]. Chronic coughs predispose to frequent increases in intra-abdominal pressure which in turn leads to weakness of the pelvic floor muscles and other supporting structures [33]. This finding however contrasts with findings in another study [32] where chronic cough was not linked to urinary incontinence.

The present study showed a significant association between level of education and urinary incontinence (p = 0.04). Women with education beyond the basic level were less likely to experience urinary incontinence compared to women with education up to the basic level. The apparent protective effect of higher education may be two-fold. The 2014 Ghana Demographic and Health Survey shows that the fertility rate is inversely related to women's educational attainment, decreasing from 6.2 among women with no education to 2.6 among women with secondary education or higher [34]. Secondly, women with lower education are more likely to engage in manual work like trading and farming that involve heavy lifting which causes constant rise in intra-abdominal pressure compared to those with higher education. Recurrent vaginal delivery and constant increase in intra-abdominal pressure may result in potentially irreversible anatomic and functional changes in the pelvic floor support structures leading to hypermobility and increased risk of stress incontinence. This finding differed from that of a Turkish study where educational status of women was not independently associated with urinary incontinence [30].

Parity was not associated with incontinence in the present study. This is in agreement with some studies [28] but inconsistent with others [27, 33].

The study findings are based on self-reported responses which could be a limitation. However, there is confidence in the results on grounds that affected women would be forthcoming with information on their affliction with urinary incontinence. It is also possible the questionnaire underestimated the burden of urinary incontinence as it has not been validated previously in the Ghanaian population as done elsewhere [35, 36]. Any bias emanating from the different routes of questionnaire administration is considered negligible as the vast majority were orally administered and is not expected to adversely affect the study findings. Policy and advocacy measures aimed at educating women and the general population on the potential causes, prevention and treatment of non-fistulous urinary incontinence are needed. Also, women should be encouraged to report the condition to clinicians when affected. Further studies are needed to validate the ICIQ-SF tool against urodynamic studies in Ghana and to assess the health seeking behaviour of women with urinary incontinence.

## Supporting information

**S1 Table. Prevalence of urinary incontinence questionnaire.**
(DOCX)

**S1 File. ICIQ-UI short form.**
(PDF)

**S2 File. Prevalence of incontinence dataset.**
(DTA)

## Acknowledgments

We are grateful to the study women for availing themselves. Many thanks to the midwives who helped with the recruitment and questionnaire administration and the rest of the study team.

## Author Contributions

**Conceptualization:** Anthony Amanfo Ofori, Henry Sakyi Opare-Addo.

**Data curation:** Anthony Amanfo Ofori, Wisdom Klutse Azanu.

**Formal analysis:** Anthony Amanfo Ofori, Joseph Osarfo.

**Methodology:** Anthony Amanfo Ofori.

**Supervision:** Anthony Amanfo Ofori.

**Validation:** Anthony Amanfo Ofori, Joseph Osarfo, Evans Kofi Agbeno.

**Writing – original draft:** Anthony Amanfo Ofori, Joseph Osarfo, Evans Kofi Agbeno, Henry Sakyi Opare-Addo.

**Writing – review & editing:** Anthony Amanfo Ofori, Joseph Osarfo, Evans Kofi Agbeno, Wisdom Klutse Azanu, Henry Sakyi Opare-Addo.

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
