## [Decision Letter · Decision Letter 0]

7 May 2020

PONE-D-20-00947

Prevalence and determinants of non-fistulous urinary incontinence among Ghanaian women seeking gynaecologic care at a Teaching Hospital.

PLOS ONE

Dear dr Ofori,

Thank you for submitting your manuscript to PLOS ONE. After careful consideration, we feel that it has merit but does not fully meet PLOS ONE’s publication criteria as it currently stands. Therefore, we invite you to submit a revised version of the manuscript that addresses the points raised during the review process.

Referring to lines 71 to 73, can you rephrase? I think that I understand that you mean: (NF)UI is more studied in industrialized countries but I may be wrong.

We would appreciate receiving your revised manuscript by Jun 21 2020 11:59PM. To enhance the reproducibility of your results, we recommend that if applicable you deposit your laboratory protocols in protocols.io, where a protocol can be assigned its own identifier (DOI) such that it can be cited independently in the future. For instructions see: http://journals.plos.org/plosone/s/submission-guidelines#loc-laboratory-protocols

We look forward to receiving your revised manuscript.

Kind regards,

Peter F.W.M. Rosier, M.D. PhD

Academic Editor

PLOS ONE

Journal Requirements:

2. In your Methods section, please provide additional information about the participant recruitment method and the demographic details of your participants.

Please ensure you have provided sufficient details to replicate the analyses such as:

a) the recruitment date range (month and year),

b) a description of any inclusion/exclusion criteria that were applied to participant recruitment and

c) a description of how participants were recruited.

3. Thank you for including your ethics statement:

'Ethical clearance was obtained from the Committee for Human Research, Publication and Ethics (CHRPE) of the Kwame Nkrumah University of Science and Technology (CHRPE/AP/239/14).'

a. Please amend your current ethics statement to confirm that your named institutional review board or ethics committee specifically approved this study.

Reviewers' comments:

Reviewer's Responses to Questions

**Comments to the Author**

1. Is the manuscript technically sound, and do the data support the conclusions?

Reviewer #1: Yes

Reviewer #2: Partly

2. Has the statistical analysis been performed appropriately and rigorously? 

Reviewer #1: I Don't Know

Reviewer #2: Yes

3. Have the authors made all data underlying the findings in their manuscript fully available?

Reviewer #1: Yes

Reviewer #2: Yes

4. Is the manuscript presented in an intelligible fashion and written in standard English?

Reviewer #1: Yes

Reviewer #2: No

5. Review Comments to the Author

Reviewer #1: Thank you for submitting this manuscript

- Please change urge incontinence to urgency incontinence

- No need to add a question to the validated questionnaire and you can simply put it in the exclusion criteria of patients.

- Why should you put the term non-fistulous in the title? You can actually mention the prevalence of urinary incontinence in the Ghana hospital.

Reviewer #2: Methodology- highlight the inclusion criteria

Results- elaborate on the impact of urinary incontinence on the lifestyle, there should be a table for it

Reference- Reference 18 should be written properly

6. PLOS authors have the option to publish the peer review history of their article (what does this mean?). If published, this will include your full peer review and any attached files.

Reviewer #1: No

Reviewer #2: No

---

## [Author Response · Author response to Decision Letter 0]

2 Jun 2020

EDITORS/REVIEWERS COMMENTS AND RESPONSES

ACADEMIC EDITOR

1, Referring to lines 71 to 73, can you rephrase? I think that I understand that you mean: (NF)UI is more studied in industrialized countries but I may be wrong.

Authors response: we have rephrased the statements to remove all ambiguities. The statement now reads “Non-fistulous urinary incontinence has been studied extensively in Europe, China and United States of America”. 

2, Please ensure that your manuscript meets PLOS ONE's style requirements, including those for file naming.

Authors response: All supporting files have been named according to the editorial requirements of PLOS ONE’s journal as

“S1_Fig.tif

 S2_Fig.tif

 S3_Fig.tif

 S1_Table.docx

 S1_File.pdf

 S2_File.dta”

 3, In your Methods section, please provide additional information about the participant recruitment method and the demographic details of your participants.

Please ensure you have provided sufficient details to replicate the analyses such as:

a) the recruitment date range (month and year),

b) a description of any inclusion/exclusion criteria that were applied to participant recruitment and

c) a description of how participants were recruited.

Authors response: Additional details about recruitment period, inclusion and exclusion criteria and recruitment procedures have been provided in the methodology section of the manuscript. The following paragraph;

“All women attending the gynaecology clinic during the study period were assessed for eligibility by midwives who were research assistants and trained on questionnaire administration with its translation into the local dialect (Asante Twi). Study objectives were explained to participants and written informed consent obtained. Exclusion criteria comprised i) declining to give consent, ii) age below 18 years, iii) pregnancy at presentation, iv) women in the puerperium during the study period and v) women with a referral diagnosis of fistulous incontinence during the study period”.

as written in the original manuscript has been rewritten in the revised manuscript as

“Between 1st January 2015 and 31st March 2015, all women accessing care at the gynaecology clinic irrespective of presenting complaints were assessed for eligibility by research assistants who were midwives working at the gynaecology clinic. Women below 18 years of age, women in their puerperium and those with a referral diagnosis of fistulous incontinence were excluded from the study at this point. The objectives of the study were explained to those eligible to partake in the study prior to consulting their physicians by the research assistants who have been trained to administer the questionnaires in English and the local (Asante Twi) language. Written informed consent were obtained from those willing to participate in the study and the questionnaire administered to them by the research assistants 

Inclusion criteria comprised All women who assessed care at the gynaecology clinic irrespective of presenting complaints during the study period and consented to participate in the study.

Exclusion criteria comprised i) women declining to give consent ii) women below 18 years of age iii) women with confirmed pregnancy during the study period iv) women in their puerperium during the study period v) women with a referral diagnosis of fistulous incontinence during the study period vi) women responding ‘NO’ to the screening question “have you experienced involuntary leakage of urine in the past one month” were excluded from completing the ICIQ-SF(the second part of the questionnaire).”

4. Thank you for including your ethics statement:

'Ethical clearance was obtained from the Committee for Human Research, Publication and Ethics (CHRPE) of the Kwame Nkrumah University of Science and Technology (CHRPE/AP/239/14).'

a. Please amend your current ethics statement to confirm that your named institutional review board or ethics committee specifically approved this study.

Authors response: The statement has been amended to read “The study was approved by the committee for Human Research, Publication and Ethics (CHRPE) of the Kwame Nkrumah University of Science and Technology (CHRPE/AP/239/14)” and same has been added to the “Ethics Statement” field of the submission form.

5, Please include captions for your Supporting Information files at the end of your manuscript, and update any in-text citations to match accordingly. 

Authors response: Caption for supporting information files has been added at the end of the manuscript and all supporting information files listed in accordance with the editorial requirements of PLOS ONE as shown below. In addition, all in-text citations have been updated to match accordingly.

Supporting information

S1 Fig. Profile of participant recruitment

S2 Fig. Referral diagnoses of study participants (%).

S3 Fig. Proportion of incontinent women by severity of incontinence (ICIQ score)

S1 Table. Prevalence of urinary incontinence questionnaire

S1 File. ICIQ-UI Short Form

S2 File. prevalence of incontinence dataset

REVIEWER #1

 Thank you for submitting this manuscript

1. Please change urge incontinence to urgency incontinence

Authors response: Urge Incontinence has been changed to urgency incontinence in the manuscript.

2. No need to add a question to the validated questionnaire and you can simply put it in the exclusion criteria of patients.

Authors response: The point is noted. The manuscript has been revised with an addition to the exclusion criteria that states 

“any participant that responded NO to the question {have you experienced any involuntary loss of urine in the past one month?} was excluded from completing the ICIQ-SF”

Subsequent to this modification, the phrase “modified version of ICIQ-SF”

 has been revised to “ICIQ-SF”.

3. Why should you put the term non-fistulous in the title? You can actually mention the prevalence of urinary incontinence in the Ghana hospital.

Authors response: The term non-fistulous was added to highlight the point that the manuscript is not about fistulous incontinence which is more associated with obstructed labour and has been studied extensively in Ghana. We are of the opinion that using the term “non-fistulous” in the title will make the study easily identifiable if one is looking for such studies in Africa or Ghana. 

REVIEWER #2

1. Methodology- highlight the inclusion criteria

Authors response: As suggested the manuscript has been revised to highlight the inclusion criteria with the inclusion of the statement

“Inclusion criteria comprised; All women who assessed care at the gynaecology clinic irrespective of presenting complaints during the study period and consented to participate in the study”.

2. Results- elaborate on the impact of urinary incontinence on the lifestyle, there should be a table for it

Authors response: The statement “On the overall interference with everyday life scale which ranged from 0 to 10 (with 0 representing no inconvenience and 10 representing a great deal of inconvenience), more than a half of the incontinent women (28/48, 58.3%) gave a score of 5 or more with 2% (1/48) giving a score of 0” has been reworded to read 

“On the overall interference with everyday life scale which ranged from 0 to 10 (with 0 representing no inconvenience and 10 representing a great deal of inconvenience) only 2% (1/48) of incontinent women reported no interference with daily life. Approximately 31% (15/48) reported mild interference with daily life whiles about 42% of incontinent women said urinary incontinence moderately interfered with their daily life”. In addition, Table 2 with the caption “interference of urinary incontinence with everyday life of affected women.” has been inserted.

3. Reference- Reference 18 should be written properly

Authors response: Reference 18 has been revised accordingly. It now reads

“Bello OO (2018) Prevalence of Non-fistulous Urinary Incontinence Among Nonparturient Women in A Tertiary Hospital. Journal of Woman's Reproductive Health 2: 35”

 instead of 

“Bello OO. JOURNAL OF WOMAN’S REPRODUCTIVE HEALTH”. as was contained in the original manuscript.

---

## [Decision Letter · Decision Letter 1]

13 Jul 2020

PONE-D-20-00947R1

Prevalence and determinants of non-fistulous urinary incontinence among Ghanaian women seeking gynaecologic care at a Teaching Hospital.

PLOS ONE

Dear Dr. Ofori,

Thank you for submitting your manuscript to PLOS ONE. After careful consideration, we feel that it has merit but does not fully meet PLOS ONE’s publication criteria as it currently stands. Therefore, we invite you to submit a revised version of the manuscript that addresses the points raised during the review process.

ACADEMIC EDITOR:

As you can see the review suggests some more changes. I would add that it is good to make clear in the discussion paragraphs that it is about women who visited the gynecological clinic. Sometimes the conclusions seem to generalize too much, as if it is a population screening.

We look forward to receiving your revised manuscript.

Kind regards,

Peter F.W.M. Rosier, M.D. PhD

Academic Editor

PLOS ONE

Reviewers' comments:

Reviewer's Responses to Questions

**Comments to the Author**

1. If the authors have adequately addressed your comments raised in a previous round of review and you feel that this manuscript is now acceptable for publication, you may indicate that here to bypass the “Comments to the Author” section, enter your conflict of interest statement in the “Confidential to Editor” section, and submit your "Accept" recommendation.

Reviewer #2: All comments have been addressed

Reviewer #3: (No Response)

2. Is the manuscript technically sound, and do the data support the conclusions?

Reviewer #2: Yes

Reviewer #3: Partly

3. Has the statistical analysis been performed appropriately and rigorously? 

Reviewer #2: Yes

Reviewer #3: I Don't Know

4. Have the authors made all data underlying the findings in their manuscript fully available?

Reviewer #2: Yes

Reviewer #3: Yes

5. Is the manuscript presented in an intelligible fashion and written in standard English?

Reviewer #2: Yes

Reviewer #3: Yes

6. Review Comments to the Author

Reviewer #2: This manuscript can be published in its present form. all the suggested corrections have been implemented.

Reviewer #3: Thank you for this contribution. This is a very appropriate study with a lack of epidemiological data in Africa.

Comments:

Major:

Materials and Methods: Line 137 and line 159: Author should consider to comment on the validity of the questionnaires used as a measuring instrument and if the questionnaires were translated into Asanti Twe language. Also to inform the reader the reason why orally administered questionnaire was used versus written answered questionnaires.

Discussion:

Line 357: Author claims parity was not associated with urinary incontinence but Table 4 indicates all of the population is parous. This will needs explanation

Line 359: The author may consider to comment on limitations or strenghts in the study:

i) if local validated questionnaire was used and

ii) written and self completed questionnaire versus orally administered questionnaire

Conclusion:

Line 365:The author should indicate if the assumption to educate clinicians re screening for urinary incontinence is valid from this study or if it still requires further research

Minor:Line 170: Spelling

References:

1. To consider to use the most lately published article on Standardization of terminology

31. This reference appears to be incomplete ... Could not access properly via Google scholar.

7. PLOS authors have the option to publish the peer review history of their article (what does this mean?). If published, this will include your full peer review and any attached files.

Reviewer #2: No

Reviewer #3: No

---

## [Author Response · Author response to Decision Letter 1]

17 Jul 2020

RESPONSES TO REVIEWERS

• ACADEMIC EDITOR

1. Academic made the following comment;

“I would add that it is good to make clear in the discussion paragraphs that it is about women who visited the gynaecological clinic. Sometimes the conclusions seem to generalize too much, as if it is a population screening.”

Response: The statement in the first paragraph of the discussion segment has been reworded to make it clear. The original statement read;

“This cross-sectional study was conducted to assess the prevalence of urinary incontinence among women, factors associated with its occurrence as well as its self-reported interference with everyday life activities of those affected” (line 289-291).

It now reads;

“This cross-sectional study was conducted to assess the prevalence of urinary incontinence among women visiting an out-patient gynaecologic clinic, determine factors associated with its occurrence as well as its self-reported interference with everyday life activities of those affected”

(line 288-291)

• REVIEWER 3

MAJOR

1. Reviewer 3 made the following extracted comment on line 137

“………and if the questionnaires were translated into Asanti Twe language. Also to inform the reader the reason why orally administered questionnaire was used versus written answered questionnaires.”

Response : it has been clarified that the bulk of the questionnaires had to be orally administered by verbal translation from English into the local language because most respondents had little or no English reading and comprehension abilities. Only a handful were able to self-administer the English version but we cannot provide data on how many could self-administer since we did not count. It was not considered relevant to the work.

The portions of the manuscript reflecting the changes are indicated below……………..

Lines 136-139 originally read as;

“physicians by the research assistants who have been trained to administer the questionnaires in English and the local (Asante Twi) language. Written informed consent were obtained from those willing to participate in the study and the questionnaire administered by the research assistants”

It now reads as (136-141);

“physicians by research assistants who had been trained to administer the questionnaires. The vast majority of questionnaires were administered through verbal translation into the local Asante Twi language as most participants had little or no English reading and comprehension skills while only a handful were self-administered in English. Written informed consent were obtained from those willing to participate in the study.” 

2. Reviewer 3 made the following comment on line 159

Author should consider to comment on the validity of the questionnaires used as a measuring instrument

Response: The questionnaire used (ICIQ-SF) has not been validated in the Ghanaian population. This would require a comparison of its output with urodynamic studies as a gold standard test with measurement of certain metrics. As earlier stated, this has not been done in the Ghanaian population and our study did not include it.

This has been clarified in the methodology and in the discussion

The portions of the manuscript reflecting the changes are indicated below……………..

Line 159-160 originally read as;

“This validated symptom questionnaire is made up of three scored questions…………………”

It now reads as (lines 161-162);

“This symptom questionnaire is made up of three scored questions………………..”

In addition, a new sentence has been introduced (lines 173-174) as;

“The ICIQ-SF, as used, has not been validated in the Ghanaian population.”

Furthermore, under the Discussion, the use of a measuring tool that has not been validated in the population has been acknowledged as a limitation and also put up for further research.

A new sentence has been introduced in the last paragraph of the DISCUSSION (line 362-364): “It is also possible the questionnaire underestimated the burden of urinary incontinence as it has not been validated previously in the Ghanaian population as done elsewhere”

ALSO, lines 366-367 originally reading as: “Further studies are needed to assess the health seeking behaviour of women with urinary incontinence” has now been modified to read as;

(Lines 370-372) “Further studies are needed to validate the ICIQ-SF tool against urodynamic studies in Ghana and to assess the health seeking behaviour of women with urinary incontinence”

3. Reviewer 3 made the following comment on line 357

 Author claims parity was not associated with urinary incontinence but Table 4 indicates all of the population is parous. This will need explanation

Response: All the population was NOT parous as Table 1 shows 115 respondents with zero parity. Table 4 only shows Odds ratios and p-values for association (parity versus incontinence) for the parity groups 1-4 and ≥5 compared to the baseline of zero parity. We think therefore that line 357 should stand.

4. Reviewer 3 made the following comment on line 359

 The author may consider to comment on limitations or strengths in the study:

i) if local validated questionnaire was used and

ii) written and self completed questionnaire versus orally administered questionnaire

Response: 

i) Issues on the validity of the questionnaire and its implications have been addressed under No. 2

ii) The question of self-administered questionnaire versus orally administered questionnaire has been addressed under No. 1. Since the vast majority of questionnaires were orally administered, limitations of selection bias arising from any disparity between the two routes is considered insignificant and will not impact negatively on the findings

 A new sentence is introduced (lines 364-367) to reflect this stance as below;

“Any bias emanating from the different routes of questionnaire administration is considered negligible as the vast majority were orally administered and is not expected to adversely affect the study findings”

5. Reviewer 3 made the following comment on line 365

The author should indicate if the assumption to educate clinicians re screening for urinary incontinence is valid from this study or if it still requires further research

Response: We have elected to do away with the recommendation in question.

 Hence, lines 365-366 which originally read as;

“Policy guidelines should aim at educating clinicians to screen for the condition in women attending gynaecology clinics” has been eliminated.

MINOR

1. The reviewer draws attention to a spelling error on line 170

 Response: 

 The misspelt word “screnning” has been changed to “screening” and is now on line 172

2. Reviewer 3 made the following comments on Reference 1. 

“To consider to use the most lately published article on Standardization of terminology”

Response:

We have replaced the old reference

 “Abrams P, Cardozo L, Fall M, Griffiths D, Rosier P, et al. (2003) The standardisation of terminology in lower urinary tract function: report from the standardisation sub-committee of the International Continence Society. Urology 61: 37-49.”

 with the most lately published article on standardization of terminology. 

“Haylen BT, De Ridder D, Freeman RM, Swift SE, Berghmans B, et al. (2010) An International Urogynecological Association (IUGA)/International Continence Society (ICS) joint report on the terminology for female pelvic floor dysfunction. Neurourology and Urodynamics: Official Journal of the International Continence Society 29: 4-20.”

The reference list has been updated appropriately.

3. Reviewer 3 made the following comments on reference 31. 

“This reference appears to be incomplete ... Could not access properly via Google scholar”.

Response:

We concede the point that the reference as appears in Google Scholar is incomplete. Unfortunately, we are not in a position to provide the additional information to make the reference complete at the moment. As a result, the old reference 

 “ Cardozo L (1997) Urogynecology: the King's approach: WB Saunders Company.” has been replaced with a more recent referrence; 

 “Gomelsky A, Dmochowski RR (2011) Urinary incontinence in the aging female: etiology, pathophysiology and treatment options. Aging Health 7: 79-88”.

 Subsequently the wording of that part of the manuscript has been amended as follows;

 Line 331-334 originally read as;

 “It has been reported that aging predisposes individuals to urinary incontinence due to reduction in bladder capacity, decreased urethral closure pressure and decreased ability to delay voiding”…

 It now reads (line 331-335)

 “The aged may be prone to the development of urinary incontinence due to hypoestrogenism, decreased urethral closure pressure and the development of urodynamic detrusor overactivity... In addition impaired mobility and increased nocturnal urine production may contribute to the development of urinary incontinence”…

Thnak you

---

## [Decision Letter · Decision Letter 2]

27 Jul 2020

PONE-D-20-00947R2

Prevalence and determinants of non-fistulous urinary incontinence among Ghanaian women seeking gynaecologic care at a Teaching Hospital.

PLOS ONE

Dear Dr. Ofori,

Thank you for submitting your manuscript to PLOS ONE. After careful consideration, we feel that it has merit but does not fully meet PLOS ONE’s publication criteria as it currently stands. Therefore, we invite you to submit a revised version of the manuscript that addresses the points raised during the review process.

ACADEMIC EDITOR:

In agreement with the reviewers comments: Can you be slightly more precise in your methods, and adapt the abstract?

We look forward to receiving your revised manuscript.

Kind regards,

Peter F.W.M. Rosier, M.D. PhD

Academic Editor

PLOS ONE

Reviewers' comments:

Reviewer's Responses to Questions

**Comments to the Author**

1. If the authors have adequately addressed your comments raised in a previous round of review and you feel that this manuscript is now acceptable for publication, you may indicate that here to bypass the “Comments to the Author” section, enter your conflict of interest statement in the “Confidential to Editor” section, and submit your "Accept" recommendation.

Reviewer #3: All comments have been addressed

2. Is the manuscript technically sound, and do the data support the conclusions?

Reviewer #3: Partly

3. Has the statistical analysis been performed appropriately and rigorously? 

Reviewer #3: I Don't Know

4. Have the authors made all data underlying the findings in their manuscript fully available?

Reviewer #3: (No Response)

5. Is the manuscript presented in an intelligible fashion and written in standard English?

Reviewer #3: (No Response)

6. Review Comments to the Author

Reviewer #3: Thank you for your response.

You have answered the questions and made some changes to the text, but some of the changes does not reflect in your abstract. This needs to be corrected

I feel the reader needs to be clearly informed in the methodology section if the ICIQ-SF was translated prior to the interview to Ghanian language but not validated, or if the person who introduced the questionnaire translated the questionnaire into Ghanian language at the same time of the interview.

7. PLOS authors have the option to publish the peer review history of their article (what does this mean?). If published, this will include your full peer review and any attached files.

Reviewer #3: No

---

## [Author Response · Author response to Decision Letter 2]

27 Jul 2020

RESPONSE TO REVIEWER No. 3

Reviewer #3: Thank you for your response.

You have answered the questions and made some changes to the text, but some of the changes does not reflect in your abstract. This needs to be corrected

I feel the reader needs to be clearly informed in the methodology section if the ICIQ-SF was translated prior to the interview to Ghanaian language but not validated, or if the person who introduced the questionnaire translated the questionnaire into Ghanaian language at the same time of the interview.

Response:

The authors appreciate the reviewer’s opinion and keen sense of detail in reviewing our work. Under Methods, It has been clarified that translation into the local language was done at the same time as the interview was being conducted. 

The relevant section of the Methods section originally read as (lines 137-141);

“……..The vast majority of questionnaires were administered through verbal translation into the local Asante Twi language as most participants had little or no English reading and comprehension skills while only a handful were self-administered in English. Written informed consent were obtained from those willing to participate in the study. “

It has now been revised to read as (139-144);

“…….The vast majority of questionnaires were administered through verbal translation into the local Asante Twi language as most participants had little or no English reading and comprehension skills while only a handful were self-administered in English. Translation of the questionnaire into the local language was done at the same time of the interview. Written informed consent were obtained from those willing to participate in the study.” 

Additionally, relevant changes have also been effected in the abstract to reflect reviewer comments.

The abstract originally read as shown below;

The study assessed the prevalence and determinants of non-fistulous urinary incontinence among gynaecologic care seekers as well as its interference with everyday life activities of affected women.

A cross-sectional study involving 400 women was conducted in a tertiary facility. Urinary incontinence was assessed using the International Consultation on Incontinence Questionnaire-short form (ICIQ-SF). The data was analysed for proportions and associations between selected variables.

The prevalence of urinary incontinence was 12%, the common types being urgency (33.3%), stress (22.9%), and mixed (20.8%). Age ≥60 years compared to 18-39 years (OR 3.66 95%CI 1.48-9.00 P=0.005), and a history of chronic cough (OR 3.80 95% CI 1.36-10.58 P=0.01) were associated with urinary incontinence. Women with education beyond the basic level were 72% less likely to experience urinary incontinence (OR 0.28 95%CI 0.08-0.96 P=0.04). Urinary incontinence interferes with everyday life activities of most affected women. 

Non-fistulous urinary incontinence is relatively common among gynaecologic care seekers yet very few women were referred with such a diagnosis. Physicians should be educated to screen all women seeking gynaecologic care for incontinence with the aim of identifying women with such conditions for appropriate treatment. 

It has now been revised to read as;

The study assessed the prevalence and determinants of non-fistulous urinary incontinence among gynaecologic care seekers as well as its interference with everyday life activities of affected women.

A cross-sectional study involving 400 women was conducted in a tertiary facility in Ghana. Urinary incontinence was assessed using the International Consultation on Incontinence Questionnaire-short form (ICIQ-SF) which has not been validated locally. The questionnaire was administered mostly in the Asante Twi language with translation done at the time of the interview. The data was analysed for proportions and associations between selected variables.

The prevalence of urinary incontinence was 12%, the common types being urgency (33.3%), stress (22.9%), and mixed (20.8%). Age ≥60 years compared to 18-39 years (OR 3.66 95%CI 1.48-9.00 P=0.005), and a history of chronic cough (OR 3.80 95% CI 1.36-10.58 P=0.01) were associated with urinary incontinence. Women with education beyond the basic level were 72% less likely to experience urinary incontinence (OR 0.28 95%CI 0.08-0.96 P=0.04). Urinary incontinence interferes with everyday life activities of most affected women. 

Non-fistulous urinary incontinence is relatively common among gynaecologic care seekers yet very few women were referred with such a diagnosis. Advocacy measures aimed at urging affected women to report the condition and educating the general population on potential causes, prevention and treatment are needed.

RESPONSE TO ACADEMIC EDITOR

In agreement with the reviewer’s comments: Can you be slightly more precise in your methods, and adapt the abstract?

Response: The detail suggested by the reviewer has been included and the abstract revised to reflect all changes made to the body of the manuscript.

The relevant section of the Methods section originally read as (lines 137-141);

“……..The vast majority of questionnaires were administered through verbal translation into the local Asante Twi language as most participants had little or no English reading and comprehension skills while only a handful were self-administered in English. Written informed consent were obtained from those willing to participate in the study. “

It has now been revised to read as (139-144);

“…….The vast majority of questionnaires were administered through verbal translation into the local Asante Twi language as most participants had little or no English reading and comprehension skills while only a handful were self-administered in English. Translation of the questionnaire into the local language was done at the same time of the interview. Written informed consent were obtained from those willing to participate in the study.” 

Additionally, relevant changes have also been effected in the abstract to reflect reviewer comments.

The abstract originally read as shown below;

The study assessed the prevalence and determinants of non-fistulous urinary incontinence among gynaecologic care seekers as well as its interference with everyday life activities of affected women.

A cross-sectional study involving 400 women was conducted in a tertiary facility. Urinary incontinence was assessed using the International Consultation on Incontinence Questionnaire-short form (ICIQ-SF). The data was analysed for proportions and associations between selected variables.

The prevalence of urinary incontinence was 12%, the common types being urgency (33.3%), stress (22.9%), and mixed (20.8%). Age ≥60 years compared to 18-39 years (OR 3.66 95%CI 1.48-9.00 P=0.005), and a history of chronic cough (OR 3.80 95% CI 1.36-10.58 P=0.01) were associated with urinary incontinence. Women with education beyond the basic level were 72% less likely to experience urinary incontinence (OR 0.28 95%CI 0.08-0.96 P=0.04). Urinary incontinence interferes with everyday life activities of most affected women. 

Non-fistulous urinary incontinence is relatively common among gynaecologic care seekers yet very few women were referred with such a diagnosis. Physicians should be educated to screen all women seeking gynaecologic care for incontinence with the aim of identifying women with such conditions for appropriate treatment. 

It has now been revised to read as;

The study assessed the prevalence and determinants of non-fistulous urinary incontinence among gynaecologic care seekers as well as its interference with everyday life activities of affected women.

A cross-sectional study involving 400 women was conducted in a tertiary facility in Ghana. Urinary incontinence was assessed using the International Consultation on Incontinence Questionnaire-short form (ICIQ-SF) which has not been validated locally. The questionnaire was administered mostly in the Asante Twi language with translation done at the time of the interview. The data was analysed for proportions and associations between selected variables.

The prevalence of urinary incontinence was 12%, the common types being urgency (33.3%), stress (22.9%), and mixed (20.8%). Age ≥60 years compared to 18-39 years (OR 3.66 95%CI 1.48-9.00 P=0.005), and a history of chronic cough (OR 3.80 95% CI 1.36-10.58 P=0.01) were associated with urinary incontinence. Women with education beyond the basic level were 72% less likely to experience urinary incontinence (OR 0.28 95%CI 0.08-0.96 P=0.04). Urinary incontinence interferes with everyday life activities of most affected women. 

Non-fistulous urinary incontinence is relatively common among gynaecologic care seekers yet very few women were referred with such a diagnosis. Advocacy measures aimed at urging affected women to report the condition and educating the general population on potential causes, prevention and treatment are needed.

---

## [Editor Report · Decision Letter 3]

29 Jul 2020

Prevalence and determinants of non-fistulous urinary incontinence among Ghanaian women seeking gynaecologic care at a Teaching Hospital.

PONE-D-20-00947R3

Dear Dr. Ofori,

We’re pleased to inform you that your manuscript has been judged scientifically suitable for publication and will be formally accepted for publication once it meets all outstanding technical requirements.

Kind regards,

Peter F.W.M. Rosier, M.D. PhD

Academic Editor

PLOS ONE

Additional Editor Comments (optional):

none
---

## [Editor Report · Acceptance letter]

3 Aug 2020

PONE-D-20-00947R3 

Prevalence and determinants of non-fistulous urinary incontinence among Ghanaian women seeking gynaecologic care at a Teaching Hospital. 

Dear Dr. Ofori:

I'm pleased to inform you that your manuscript has been deemed suitable for publication in PLOS ONE. Congratulations! Your manuscript is now with our production department. 

Kind regards, 

on behalf of

Dr. Peter F.W.M. Rosier 

Academic Editor

PLOS ONE